# The Missing Alignment Link of In-context Learning on Sequences

**Harshvardhan Agarwal** [1]   **Sunita Sarawagi** [1]

## Abstract

Large language models (LLMs) have demonstrated the capability to perform in-context learning (ICL) for completely unseen tasks in classification or language completion. Sequence to sequence (Seq2Seq) is another popular task category with several applications seeking quick adaptation with ICL. We present a systematic analysis of the ICL capability of LLMs on Seq2Seq tasks using a formal structured language-pair. Our study reveals a critical limitation: except for very short input sequences, ICL fails to achieve consistent learning across all output positions. This exposes a fundamental weakness of modern LLMs — their inability to effectively uncover the alignment between input and output sequences. Consequently, this limitation results in incomplete induction heads, which form the basis for in-context learning of new discrete mappings.

To address these limitations, we propose ICA-Tune, a method for focused fine-tuning of an LLM using in-context examples. We present a mechanistic evaluation with two accuracy probes to show how alignment emerges in middle layers of an LLM without any direct supervision. This alignment leads to an abrupt jump in the completeness of the induction heads in higher layers. We show that compared to standard fine-tuning, ICA-Tune enables more sample efficient learning and generalizes better to OOD instances.

## 1. Introduction

Large Language Models (LLMs) have demonstrated the capability of In Context Learning (ICL) (Brown et al., 2020) a new task, where given $k$ input-output pairs and a test input

as a prompt $\mathbf{x}^1, \mathbf{y}^1, \ldots, \mathbf{x}^k, \mathbf{y}^k, \mathbf{x}^*$, they predict a $\hat{\mathbf{y}}$ indicative of having learned the task. The success of ICL has been demonstrated on several tasks (Bertsch et al., 2024; Agarwal et al., 2024), including novel key-value mappings (Kossen et al., 2024), and new synthetic language learning tasks (Rajaraman et al., 2024; Bietti et al., 2023; Edelman et al., 2024; Akyürek et al., 2024). For such unseen tasks, a widely accepted explanation is that ICL arises by the formation of induction circuits (Olsson et al., 2022; Reddy, 2024; Singh et al., 2024). An induction circuit requires at least two layers for formation as illustrated in Figure 1. In the first stage, each label position $i$ copies over its previous $\mathbf{x}^i$ as a key. In the second stage, the test input $\mathbf{x}^*$ is used as a query to copy over the label $\mathbf{y}^i$ ($i \leq k$) with the matching key $\mathbf{x}^i$. This explanation also works for learning new languages where the key corresponds to n-grams (Rajaraman et al., 2024; Bietti et al., 2023; Edelman et al., 2024; Akyürek et al., 2024).

In this paper, we investigate the effectiveness of in-context learning (ICL) on unseen sequence to sequence tasks. In this case each input $\mathbf{x}$ is a sequence $x_1, \ldots, x_m$ of tokens, likewise each output $\mathbf{y}$ is a sequence $y_1, \ldots, y_n$. ICL for Seq2Seq is useful in several tasks such as semantic parsing for custom APIs (Roy et al., 2023), translation from or to rare languages (Garcia et al., 2023), and Text2SQL generation for new private enterprise schema. To study the mechanism of ICL without fear of data contamination, we follow the practice in prior work of evaluating on new synthetic tasks. Our synthetic generator is inspired by real languages. For each task, we sample a probabilistic context free grammar generating $\mathbf{x}$, and sample $P(\mathbf{y}|\mathbf{x})$ using an alignment function inspired from classical models used in lexical translation (Brown et al., 1993), and a probabilistic finite state automata. An example of such a setup is shown as the first row of Table 1. Over various characteristics of these structured Seq2Seq tasks, we evaluate pre-trained open-source LLMs for their ICL capabilities.

We observe that except for very short $\mathbf{x}$ sequences, LLMs fail to in-context learn Seq2Seq tasks even when they succeed in previously studied classification and language learning tasks. We attribute the failure to the inability of the LLM to learn new x-y alignments. ICL for Seq2Seq tasks entails two steps: (1) learning input-output alignments, and (2) learning the next $y$ token given previous $y$-s and the aligned $x$. The second step can be learned by induction heads, but

[1]Computer Science and Engineering, Indian Institute of Technology Bombay, Mumbai, India. Correspondence to: Harshvardhan Agarwal <hvag976@stanford.edu>, Sunita Sarawagi <sunita@iitb.ac.in>.

*Proceedings of the 42nd International Conference on Machine Learning*, Vancouver, Canada. PMLR 267, 2025. Copyright 2025 by the author(s).

for the first step we show that modern LLMs do not learn in-context. We present a set of counterfactual experiments to demonstrate that inability to learn the alignment in-context is the reason for poor accuracy on long sequences.

Motivated by these observations, we propose ICA-Tune for focused fine-tuning of a subset of the LLM parameters in ICL mode instead of standard per-example fine-tuning. For monotonic alignments, fine-tuning just the KQ attention parameters of a single layer suffices. Using this hybrid of ICL and fine-tuning, we observe huge jump in accuracy of prediction with just a few examples. We present a mechanistic evaluation of the formation of the induction circuit via ICA-Tune. We show that even with standard next-token loss, early layers learn the input-output alignment without any direct supervision, and higher layers learn to inductively lookup labels across sequences. Also, alignment accuracy rises a few steps before lookup accuracy abruptly rises in higher layers. Previous studies have also reported abrupt emergence of induction circuits, but unlike earlier induction circuits ours involves an additional alignment learning layer across non-continuous positions to setup the context keys.

We show that ICA-Tune's hybrid approach to adaptation leads to more efficient learning than conventional fine-tuning and better OOD generalization. We conclude with a discussion of the challenges of supporting in-context learning of alignment functions.

**Contributions** (1) We design a formal language to highlight the inability of modern LLMs to in-context learn structured Seq2Seq tasks. (2) We explain that the main hindrance to ICL for Seq2Seq tasks is the inability to in-context learn task-specific input and output alignments within each example. (3) We present ICA-Tune, a hybrid of fine-tuning and ICL, where we fine-tune a few parameters with next token prediction loss. (4) We show interesting patterns of abrupt learning of a new type of induction circuit where early layers capture immediate context, middle layers learn input-output alignments, and higher layers support lookup with induction heads. [1] (5) We show that sample efficiency and OOD generalization is higher for ICA-Tune compared to standard fine-tuning.

## 2. Related Work

Ever since in-context learning (ICL) was discovered as an emergent phenomenon of LLMs trained with the next-token prediction loss (Brown et al., 2020), ICL has been extensively evaluated and analyzed. Prior work on evaluation of ICL spans various task types: regression where $\mathbf{y}$ is real (Garg et al., 2022), scalar classification where $\mathbf{y}$ is dis-

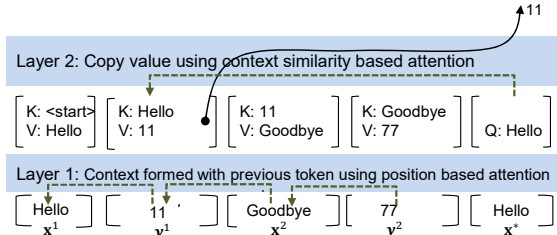

*Figure 1.* Illustration of in-context learning via induction heads for a scalar classification task with $k = 2$ examples.

crete (Kossen et al., 2024; Shi et al., 2024; Bertsch et al., 2024), language learning where $\mathbf{y}$ is a sequence from a language and $\mathbf{x}$ is empty (Edelman et al., 2024; Akyürek et al., 2024), and sequence to sequence learning where both $\mathbf{x}$ and $\mathbf{y}$ are sequences (Roy et al., 2023). Across these different studies the broad consensus is that ICL does provide accuracy gains beyond zero-shot. However, for structured Seq2Seq tasks like semantic parsing, the success of ICL crucially depends on carefully selecting examples to provide adequate coverage of the test example, and reduce interference (Levy et al., 2023). Our study provides more insights on this phenomenon.

Puzzled by the empirical success of ICL, several studies have sought to explain how pre-trained LLMs could develop ICL capabilities. We discuss in detail some of these studies in Appendix A. For totally unseen input-output mappings, a widely accepted explanation is the formation of an induction circuit (Olsson et al., 2022) as shown in Figure 1. Chen et al. (2024) shows why pre-training with next-token loss orients the transform parameters to form such induction circuits due to the presence of repeated structures naturally in the pre-training corpus. Reddy (2024) and Singh et al. (2024) study the dynamism of the formation of induction circuits during training using synthetic scalar classification datasets. For scalar classification, the context in the induction head is always the previous x-tokens input. ICL of new regular languages is studied in (Akyürek et al., 2024; Edelman et al., 2024; Rajaraman et al., 2024) with synthetic languages sampled from finite state automaton and markov chains. For this task, the context in induction head is a few tokens immediately preceding each token that define an n-gram. And thus, the induction circuits help to implement an n-gram language model, that suits Markovian languages.

We are aware of no prior work that systematically analyzes ICL on Seq2Seq learning tasks with formal structured languages. Such a study is important because many real-life tasks, example semantic parsing, low resource translation, error correction, and Text2SQL are instances of seq2seq learning task. Existing work has been limited to just overall accuracy on real datasets where it is hard to control for contamination from the training set.

---

[1] We release code for data generation and ICA-Tune at `https://github.com/draco976/icatune`

# 3. Studying ICL on Seq2Seq tasks

**Problem Formulation**   We denote an input sequence $\mathbf{x} : x_1, \ldots, x_m$ as comprising of $m$ discrete tokens, and an output sequence $\mathbf{y} : y_1, \ldots, y_n$ consisting of $n$ tokens where each $x_i, y_j \in \mathcal{V}$, a vocabulary of tokens. In general $n, m$ can vary across instances. Let $\mathcal{X}, \mathcal{Y}$ denote the input and output spaces of discrete token sequences of arbitrary lengths. Each sequence-to-sequence prediction task $\tau$ is characterized by a distribution $P_\tau(\mathcal{X})$ over input sequences and conditional distribution $P_\tau(\mathcal{Y}|\mathcal{X})$ over the output sequences. To learn this task we are given a small number $k$ of input-output sequence pairs $\{(\mathbf{x}^1, \mathbf{y}^1), \ldots (\mathbf{x}^k, \mathbf{y}^k)\}$ sampled from $P_\tau(\mathcal{X}, \mathcal{Y})$. Our goal is to explore the use of in-context-learning on a pre-trained LLM $M_\theta$ to learn this task from the given examples. In ICL, $M_\theta$ will be provided as input the $k$ examples $\{(\mathbf{x}^1, \mathbf{y}^1), \ldots (\mathbf{x}^k, \mathbf{y}^k)\}$ followed by a new test instance $\mathbf{x}^* \sim P_\tau(X)$, and $M_\theta$ needs to adapt to the new task using just one forward pass over the input.

Since $k$ is small, and we are exploring tasks that are unseen by $M_\theta$, learning $P_\tau(\mathcal{Y}|\mathcal{X})$ is only possible if $P_\tau(\mathcal{Y}|\mathcal{X})$ has a decompositional structure to allow generalization to sequences of varying length.

**Decompositional structure of $P_\tau(\mathcal{Y}|\mathcal{X})$**   As LLMs generate tokens auto-regressively, we first rewrite $P_\tau(\mathbf{y}|\mathbf{x}) = \prod_{j=1}^n P_\tau(y_j|\mathbf{x}, y_1, \ldots, y_{j-1})$. For tractable learning with small $k$, we simplify the dependence structure of $y_j$ as follows: (1) Instead of depending on all previous $y$ tokens, each $y_j$ depends on a small window of size at most $g$ over previous $y$ tokens. (2) Instead of all $\mathbf{x}$ tokens, a latent alignment function $A_\tau(j)$ identifies the sub-part of $\mathbf{x}$ relevant to predict $y_j$. Thus, the task dependent Seq2Seq model is

$$P_\tau(\mathbf{y}|\mathbf{x}) = \prod_{j=1}^n P_\tau(y_j|\mathbf{y}_{j-g:j-1}, \mathbf{x}_{A_\tau(j)}) \qquad (1)$$

In order to learn $P_\tau(\mathbf{y}|\mathbf{x})$, ICL needs to discover both the input-output alignment function $A_\tau(j)$ and the conditional token distribution $P_\tau(y_j|\mathbf{y}_{j-g:j-1}, \mathbf{x}_{A_\tau(j)})$. We study an LLM's capability for such learning using synthetic tasks that allow a systematic exploration without fear of contamination from the huge training set of LLMs. Towards this end, we define a class of formal sequence to sequence task that allows flexible controls of both the alignment function $A_\tau(j)$ and the overall $P_\tau(y_j|\mathbf{y}_{j-g:j-1}, \mathbf{x}_{A_\tau(j)})$.

## 3.1. Formal sequence to sequence generation model

For each task $\tau$, we assume $\mathcal{X}$ represents a *structured* formal language, such as a context free language, that is strictly more powerful than a regular Markovian language. Each element $x$ of $\mathbf{x}$ comes from a small discrete vocabulary $\Sigma_\tau$. Let $P_\tau(\mathbf{x})$ denote the distribution over $\mathbf{x}$, which for

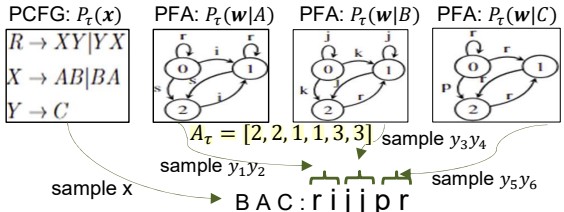

*Figure 2.* Illustration of our data generation process here $m = 3, c = 2$ and attention is non-monotonic

our experiments is represented as a simple two-level probabilistic context free grammar PCFG(X) described in the Appendix B. The characterization of the distribution $P_\tau(\mathbf{x})$ is not too critical since $\mathbf{x}$ is always given. The learning tasks is to infer $P(\mathbf{y}|\mathbf{x})$.

Our model for $P(\mathbf{y}|\mathbf{x})$ is inspired from the classical lexical translation model (Brown et al., 1993; Dyer et al., 2013). Given an $m$ length input $\mathbf{x}$, the corresponding $\mathbf{y}$ sequence consists of $m$ segments $w_1, \ldots, w_m$, each $w_p$ can be viewed as a y-phrase that aligns with a $x_q$ at position $q$ as determined by a task-specific alignment function $\tilde{A}_\tau(p) \mapsto [m]$. We next describe our method of sampling the alignment function and generating the y-phrases conditioned on the aligned $x$.

**Sampling alignment function.** We choose the task specific alignment function for the $p$-th phrase $w_p$ as:

$$\tilde{A}_\tau(p) = q, \text{ where } q \sim \frac{\exp(-\lambda|q - p|)}{\sum_{s=1}^m \exp(-\lambda|s - p|)}, p, q \in [m] \qquad (2)$$

By choosing $\lambda \to \infty$ we can make the alignment monotonic ($\tilde{A}_\tau(p) = p$), and for $\lambda \to 0$ the alignment approaches almost a random permutation. We assume here that all y-phrases are of the same length $c$, and thus the alignment function for the $j$-th y-token is $A_\tau(j) = \tilde{A}_\tau(\lfloor \frac{j}{c} \rfloor)$.

**Sampling y-phrases.** Each y-phrase $w_p$ is generated as a regular language given the aligned $x_q$ where $q = \tilde{A}_\tau(p)$. The regular language for any $x$ is represented as a probabilistic finite state automaton PFA($\Sigma_{\tau|x}, S_{\tau|x}, T_{\tau|x}$) where $\Sigma_{\tau|x}$ denotes the vocabulary, $S_{\tau|x}$ denotes the set of states in the PFA, and $T_{\tau|x}$ denotes the stochastic transition matrix. More details about PFAs, how we choose a PFA for each $x$, and how we generate strings from a PFA can be found in Appendix B. Figure 2 and Table 1 shows examples of generated $x, y$ pairs with $m = 3, c = 2$. Note in the generated $\mathbf{y}$ there is no demarcation among the y-phrases. A pseudocode for our data generator appears in Algorithm 1.

**Real-life motivation.** As a real-life example, think of $\mathbf{x}$ as a sentence in English, and $\mathbf{y}$ its translation in Japanese. The English language follows Subject-Verb-Object word order

| Prompt | Prompt string: $\mathbf{x}^1 : \mathbf{y}^1 \; \mathbf{x}^2 : \mathbf{y}^2 \; \mathbf{x}^3 : \mathbf{y}^3 \; \mathbf{x}^*$ |
|---|---|
| Standard | `ACB: rijjpr CAB: jjriwp ABC: rtprjh BCA:` |
| Pre-aligned | `ACB: AriCjjBpr CAB: CjjAriBwp ABC: ArtBprCjh BCA:` |

*Table 1.* First row marked Standard shows the prompt for ICL on a synthetic Seq2Seq task. The second row marked Pre-aligned is an oracle setup where the input and output tokens have been interleaved using knowledge of gold alignment. Here $m = 3, c = 2$ and alignment is monotonic.

---

**Algorithm 1** Sequence Generation Process

1: **Input:** $k, m, c, \lambda$
2: $\Sigma_\tau \leftarrow m$ random capital English letters.
3: $P_\tau(\mathbf{x}) \leftarrow$ Sample a PCFG($m, \Sigma_\tau$) (Sec B.1)
4: $\tilde{A}_\tau(\cdot) \leftarrow$ Sample alignment ($m, \lambda$) using Eq 2.
5: $P_\tau(\cdot|x) \leftarrow$ Sample a PFA($\Sigma_{\tau|x}, S_{\tau|x}, T_{\tau|x}$) foreach $x \in \Sigma_\tau$ (Sec B.2)
6: **for** each $i \in \{1, \dots, k+1\}$ **do**
7:    $\mathbf{x}^i \leftarrow$ Sample a sequence from PCFG $P_\tau(\mathbf{x})$
8:    $\mathbf{y}^i \leftarrow \phi$
9:    **for** each $p \in \{1, \dots, m\}$ **do**
10:       $w_p \leftarrow$ Sample $c$ tokens from PFA $P_\tau(\cdot|\mathbf{x}^i_{\tilde{A}_\tau(p)})$
11:       $\mathbf{y}^i \leftarrow \mathbf{y}^i + w_p$
12: **return** $\mathbf{x}^i, \mathbf{y}^i : i = 1, \dots k+1$

---

as against Subject-Object-Verb in Japanese. The alignment function captures this systematic difference, whereas the PFA models the Japanese character sequence corresponding to each English word.

### 3.2. Evaluating In-context Learning of Seq2Seq Tasks

We start with an evaluation of ICL on unseen Seq2Seq tasks using the above synthetic data generator.

**Model** For our experiments, we utilize the LLaMA 3 model (Meta, 2024) with 8 billion parameters. In Section 3.3 we present results with other LLMs. For each task characterized as defined above, we generate $k + 1$ input-output sequence pairs $\{(\mathbf{x}^1, \mathbf{y}^1), \dots (\mathbf{x}^{k+1}, \mathbf{y}^{k+1})\}$ using Algorithm 1. We create a prompt using the first $k(= 15)$ as in-context example, and test instance $\mathbf{x}^*$ as $\mathbf{x}^{k+1}$. We then generate the $y$ tokens step-by-step using teacher forcing where at step $j$ we predict $\hat{y}_j$ with prompt as ICL examples, $\mathbf{x}^*$, followed by $y_1^{k+1} \dots y_{j-1}^{k+1}$. We measure accuracy by checking if $\hat{y}_j$ is accepted by the true PFA.

In Figure 3(a) we present accuracy for different lengths of $\mathbf{x}$ sequences ($m$), and y-phrase ($c$) for the simplest case of monotonic alignments ($\lambda = \infty$) where each y-phrase $w_p$ aligns with $x_p$. We make a number of observations:

**ICL accuracy drops with increasing $m$.** When sequence length $m = 1$, the Seq2Seq task reduces to a regular language learning task since all tokens in a $\mathbf{y}$ sequence are sampled from a single regular grammar represented by the PFA. In this setting, as various previous studies (Edelman et al., 2024; Akyürek et al., 2024; Rajaraman et al., 2024) have shown, ICL succeeds in providing very high accuracy since induction heads can approximate regular language well using n-grams. However, as sequence length $m$ is increased the accuracy keeps dropping.

**ICL accuracy increases with increasing y-phrase length for $m > 1$.** When the length of each y-phrase is large, tokens later in each y-phrase can be determined based on previous tokens in the y-phrase, and there n-gram based matching is successful. However, a token at the start of each y-phrase critically depend on its corresponding aligned $x$ token for correct prediction. In Figure 3(b) we show accuracy for the first token within each y-phrase. For the first $y$ token in a y-phrase accuracy is significantly lower than overall accuracy because it is dependent only on the aligned $x$ token, which is separated from $y_j$ in the prompt by other irrelevant tokens[2]. We characterize this as a failure to recover the input-output alignment.

**Interleaving $\mathbf{y}$ with aligned $x$ tokens enables ICL.** We perform a counterfactual experiment in which the $\mathbf{y}$ sequence is interleaved with its aligned tokens from the $\mathbf{x}$ sequence, assuming oracle knowledge of $A_\tau()$. The second row of Table 1 shows an example of the prompt used for such pre-aligned sequences. In Figure 3(c), we report the sequence prediction accuracy using the same axes as in Figure 3(a). Remarkably, accuracy increases sharply across all settings. The interleaving causes each $y$ token to be preceded by all relevant tokens in the LLM prompt, and thereafter in-context learning is able to learn this new task via the formation of induction heads.

### 3.3. Do these observations generalize across LLMs?

To assess whether these trends persist at larger scales, we repeated the same experiment on several frontier LLMs with $m = 8$ and $c = 1$, comparing performance under the standard and pre-aligned prompting strategies. As shown in Table 2, even very large models exhibit low accuracy with the standard prompt. However, when provided with

---

[2]Since we assumed a non-Markovian distribution for $P_\tau(\mathbf{x})$, a longer context on the previous $y$ tokens cannot substitute for $\mathbf{x}_{A_\tau(j)}$.

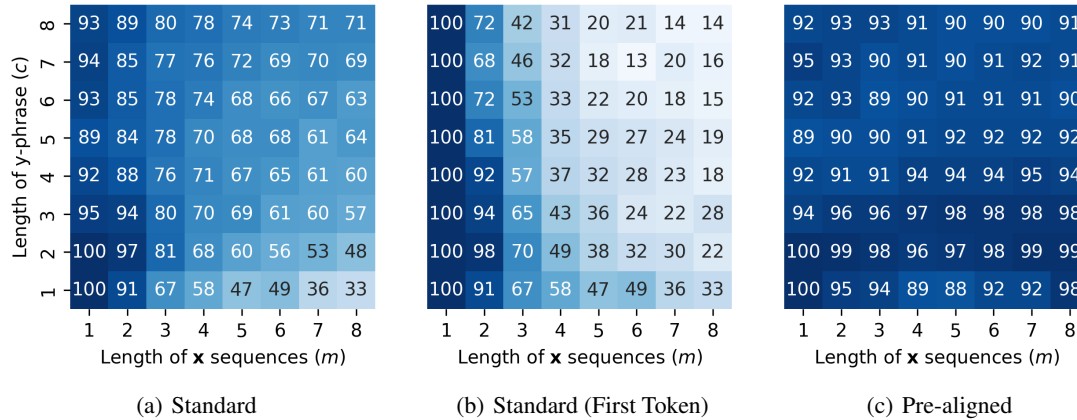

(a) Standard

(b) Standard (First Token)

(c) Pre-aligned

*Figure 3.* Studying ICL Accuracy over sequences with different lengths of y-phrase($c$) and $\mathbf{x}$ sequences ($m$) with (a) Accuracy drops with increasing sequence length and increasing y-phrase length for $m > 1$. (b) Largest drop in accuracy with $m$ is for the first token in y-phrase. (c) High accuracy overall after interleaving gold aligned $x$ token within the $\mathbf{y}$ sequence. These show that **Low accuracy for large sequences is because of lack of alignment.**

pre-aligned input sequences—where $x$ and $y$ tokens are interleaved—the accuracy improves drastically. This reinforces the conclusion that poor performance under standard prompting arises not from limited model capacity, but from misalignment in the input structure.

| Model | Standard | Pre-Aligned |
|---|---|---|
| Llama-3.3-70B[3] | 20.00 | 91.25 |
| Llama-3.1-405B[3] | 31.25 | 98.75 |
| GPT-4o | 36.25 | 100.0 |
| Claude-3.7-Sonnet | 51.25 | 82.50 |

*Table 2.* Prediction accuracy (%) for frontier LLMs under standard vs. pre-aligned prompting. Even very large models underperform with standard prompts, while alignment significantly improves outcomes.

We repeat the ICL evaluation from Section 3.2 using Llama3.2-1B and Llama3.2-3B, and compare them with Llama3.3-8B in Figure 4. Accuracy increases with larger y-phrase length ($c$) across all models, though larger models achieve higher absolute accuracy. With pre-aligned prompts (Figure 4(b)), even the smallest model performs well, demonstrating that ICL is effective when relevant $x$ tokens are placed adjacent to $y$ tokens. In contrast, with standard prompts (Figure 4(a)), all models struggle for small $c$ due to the need to infer alignments in-context.

In the above experiments we assumed oracle knowledge of the alignment function $A_\tau(j)$ in creating the aligned sequence with $x$ and $y$ tokens interleaved. In Section 5 we present a discussion of why current LLMs may not be equipped to in-context learn alignments. We therefore

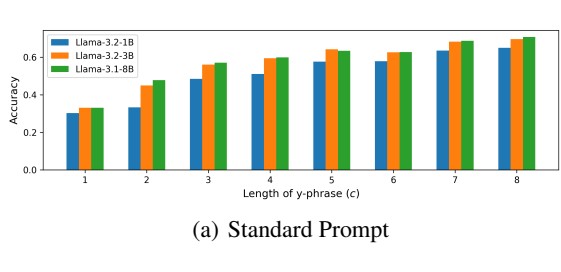

(a) Standard Prompt

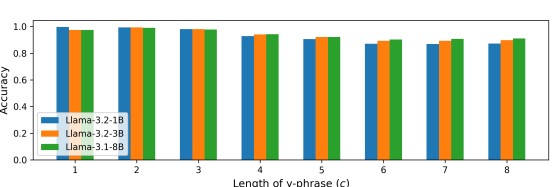

(b) Pre-aligned Prompt

*Figure 4.* Prediction accuracy for standard and aligned prompts for different lengths of y-phrase ($c$) and $m = 8$ for three different Llama model sizes.

propose additional fine-tuning to learn new Seq2Seq tasks.

## 4. Fine-tuning to learn Input-Output Alignments

Our goal is to fine-tune the LLM so as to learn the task-specific alignment function $A_\tau()$, and leverage the LLM's existing capability to lookup labels via induction heads to learn the local token distribution $P_\tau(y_j|\mathbf{y}_{j-g:j-1}, \mathbf{x}_{A_\tau(j)})$. Once $y_j$ tokens get contextualized with their aligned $x$-tokens, the higher layers can form the induction heads with keys as $[\mathbf{y}_{j-g:j-1}, \mathbf{x}_{A_\tau(j)}]$ and values as $y_j$.

Since we do not have any supervision on the alignment, we fine-tune the model to minimize next token prediction loss over tokens of $\mathbf{y}$ sequence. However, we perform this fine-

---

[3]Both Llama-3.3-70B and Llama-3.1-405B are the "Instruct-Turbo" variants.

tuning in the in-context setting where each training instance is over a set of $k + 1$ examples where the first $k$ form the in-context examples. To fine-tune for a given task $\tau$, we iteratively sample $k + 1$ random input-output sequences from $P_\tau(X, Y)$ to maximize this likelihood:

$$\max_{\theta_a} \mathbb{E}_{\{(\mathbf{x}^i, \mathbf{y}^i): i=1\ldots k+1\} \sim P_\tau} \sum_{i=1}^{k+1} \sum_{j=1}^{n} \log M_\theta \left( Z_j^i \right) [y_j^i] \quad (3)$$
$$Z_j^i = [\mathbf{x}^1, \mathbf{y}^1, \ldots \mathbf{x}^{i-1}, \mathbf{y}^{i-1}, \mathbf{x}^i, y_1^i, \ldots, y_{j-1}^i]$$

where $M_\theta(Z)$ denotes the next-token probability distribution output by a transformer in response to an input prompt $Z$. We fine-tune only the attention parameters $\theta_a$ of the LLM using LoRA (Hu et al., 2021), and call this method ICA-Tune. In the Appendix E.1 we show that in many cases fine-tuning only a select one LLM layer suffices. More details of our fine-tuning setup can be found in Appendix D.

### 4.1. Results of ICA-Tune

In Figure 5(a) we present the evolution of accuracy against fine-tuning steps for different seeds. We present both overall accuracy, and accuracy segregated by the position of a y-token in its y-phrase. We see that across all seeds, accuracy shoots up most for the first token in a y-phrase, most likely since that is the token for which the missing alignment to the $x$-token is most crucial. To understand the learning mechanism we monitor two types of intermediate accuracy along the fine-tuning process for each layer $L$ of $M_\theta$.

1. **Alignment Accuracy:** The fraction of time each $y_j^k$ has greater attention to its aligned $x$-tokens $\mathbf{x}_{A_\tau(j)}^k$ than other tokens within each example. This measures how well the model aligns $y$ tokens with their relevant $x$ tokens. We measure this alignment accuracy only for first token of each y-phrase.

$$\frac{1}{m} \sum_{t=1, t+=c}^{n} \mathbb{1} \left( \text{argmax}_j \, \alpha^L(y_t^{k+1}, x_j^{k+1}) = A_\tau(t) \right)$$

$\alpha^L(a, b)$ is the mean attention weight assigned by token $a$ to the token $b$ in attention layer $L$.

2. **IC-Lookup Accuracy:** The proportion of attention weights from each $y$ position in the test (last) sequence $\mathbf{x}^{k+1}$ to the correct $y$-tokens from previous sequences provided for in-context learning. This reflects how accurately the model looks up past outputs during generation of $\mathbf{y}^{k+1}$ and is measured as:

$$\frac{\sum_t^n \sum_i^k \sum_j^n \alpha_-^L(y_t^{k+1}, y_j^i) * \mathbb{1} \left( x_{A_\tau(t)}^{k+1} = x_{A_\tau(j)}^i \right)}{\sum_t^n \sum_i^k \sum_j^n \alpha_-^L(y_t^{k+1}, y_j^i)}$$

$\alpha_-^L(a, b)$ is attention at layer $L$ from the position where $a$ is generated to the position where token $b$ is input.

Figure 5(b) plots the alignment accuracy and Figure 5(c) plots the IC-Lookup accuracy across fine-tuning steps for different layers. These bring out many interesting patterns.

**ICA-Tune causes alignment to emerge without any direct supervision.** ICA-Tune's loss function (Eq 3) is standard next-token prediction loss with no explicit role or mention of alignment. However, by fine-tuning with in-context examples, the correct input-output alignment emerges in certain layers. This likely happens because the pre-trained Llama model has a propensity to harness induction heads for prediction with in-context examples. The alignment learning provides the induction heads with more informative keys.

**Alignment accuracy is high in middle layers, IC-Lookup high in upper layers.** As seen in Figure 5(b), alignment accuracy is high in middle layers layers 14–16. Layers above 16 and below 14 show poor alignment accuracy. IC-Lookup is more accurate in higher layers (e.g. layers 17–19) whereas layers 14–16 that have high alignment accuracy show poor IC-Lookup accuracy. For Seq2Seq tasks induction circuits requires three stages: First, lower layers establish the context — for example figuring out the inter-sequence and $\mathbf{x}, \mathbf{y}$ boundaries, and assigning relative position counts within each $\mathbf{y}$ sequence. Subsequently, the middle layer learns the alignment function $A_\tau(j)$. Finally, this leads to the formation of the induction heads in higher layers. We show a diagram illustrating a plausible three layer induction circuit in Figure 10 of the Appendix.

**Alignment accuracy emerges first abruptly, followed by an increase in IC-Lookup accuracy.** The alignment accuracy rises abruptly around step 85 in Fig 5(b), the IC-Lookup accuracy rises slightly later around step 110 in Fig 5(c). This shows that it is necessary for input-output alignments to form within an example, before a test example can lookup matching $y$ tokens from in-context examples.

We corroborate this point further by visualizing raw attentions at two different layers $\ell = 14, \ell = 18$ for the baseline model and the ICA-Tuned model. Figures 6(a) shows attention among the tokens of $\mathbf{x}^k, \mathbf{y}^k$ at layer $\ell = 14$ before and after fine-tuning for $m = 3, c = 2$. Here the $\mathbf{x}$ tokens are C A B and $\mathbf{y}$-tokens are o a s j e l. In the baseline, there is very little attention from the y-tokens to the x tokens. In the ICA-Tuned version, there is much higher attention of y-tokens to their aligned x-tokens (highlighted in red). For example, from o a we see alignment to C, and from s j we see alignment to A. Figures 6(b) examines attention from the tokens of the test sequence $\mathbf{x}^{k+1}, \mathbf{y}^{k+1}$ to the last ICL sequence $\mathbf{x}^k, \mathbf{y}^k$ at layer $\ell = 18$ before and after fine-tuning. Note here $\mathbf{x}^{k+1} = $ C A B and $\mathbf{x}^k = $ B C A. Thus, the first y-phrase of the test sequence should lookup the second y-phrase of the IC example, and so on. We observe such a pattern in the fine-tuned model's atten-

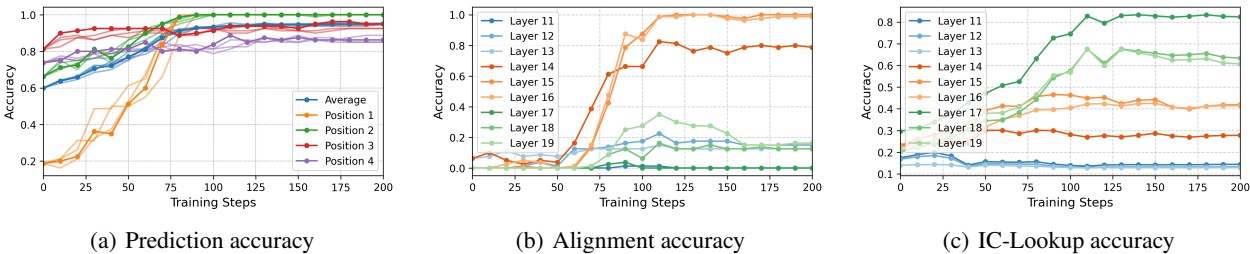

(a) Prediction accuracy   (b) Alignment accuracy   (c) IC-Lookup accuracy

*Figure 5.* Learning dynamics of ICA-Tune with increasing steps. (a) Prediction Accuracy for four different seeds: average and at different positions along the y-phrase. **Maximum jump for $y$ tokens at the start of a y-phrase.** (b) The gains are due to the emergence of $\mathbf{x}, \mathbf{y}$ alignment. The plot shows extracted alignment accuracy at different LLM layers. **Alignment accuracy high in middle layers (15,16) and abruptly rises between steps 75 and 90.** (c) **IC-Lookup ability is seen in higher layers (17, 18) above the layers where alignment emerges. This shows that alignment is needed for formation of informative induction heads.**

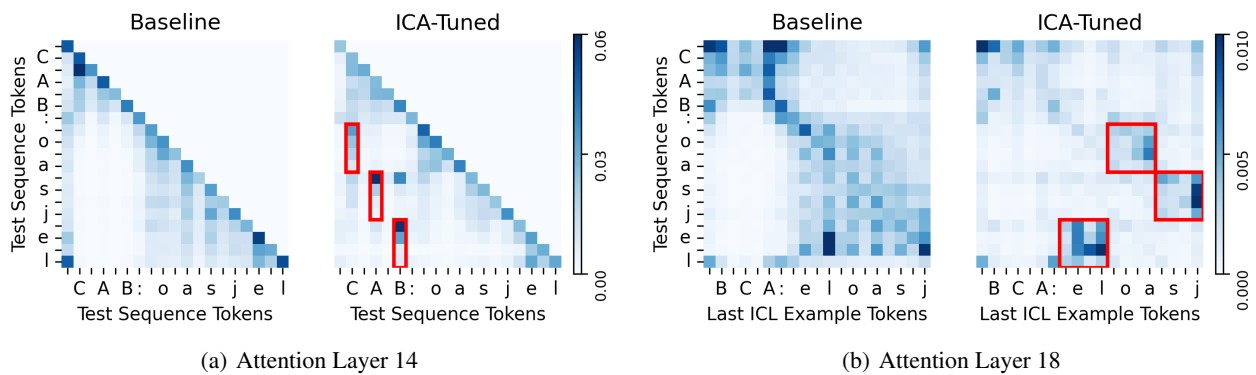

(a) Attention Layer 14   (b) Attention Layer 18

*Figure 6.* (a) Attention heatmaps for layer 14 of the baseline and fine-tuned model. The heatmaps show how attention is distributed between the y-tokens (target tokens) and the corresponding x-tokens (input tokens). In the ICA-Tuned model, initial y-tokens tend to focus on their corresponding x-tokens more consistently compared to the baseline. (b) The attention heatmaps for layer 18 between tokens from the test example (on the y-axis) and to tokens of nearest IC example (on the x-axis). Attention (shown as red squares) of test sequence `C A B` to in-context sequence `B C A` reflects the order of their x-tokens.

tion weights, but in the baseline model the attention from the $\mathbf{y}^{k+1}$ to $\mathbf{y}^k$ tokens is largely uniform. This shows that the induction heads formed with the relevant aligned context in the fine-tuned model, and did not form in the baseline.

### 4.2. ICA-Tune for another model

We reinforce the above observations on the learning dynamics of ICA-Tune by repeating the experiments of Figure 5 using a smaller Qwen2.5-3B model. The results as shown in Figure 7, reveal the same three-stage progression: (1) a sharp increase in prediction accuracy at the start of each y-phrase, (2) the emergence of alignment attention patterns in middle layers (e.g., layers 23–24), and (3) the appearance of ICL-style lookups in higher layers (e.g., layers 28–29), above the layers where alignment emerges. Despite being significantly smaller than Llama-3, the Qwen model exhibits the same inductive behavior, suggesting that the observed alignment-induced mechanism is robust across models.

### 4.3. Comparison with Standard Fine-tuning

Here we show that fine-tuning in ICL mode was essential for the alignment to emerge by comparing ICA-Tune with standard fine-tuning where we minimize loss on examples sampled independently. Thus, unlike Equation 3, in standard fine-tuning the objective becomes:

$$\max_{\theta_a} \mathbb{E}_{(\mathbf{x},\mathbf{y}) \sim P_\tau(X,Y)} \sum_{j=1}^{n} \log T_\theta(\mathbf{x}, y_1, \dots, y_{j-1})[y_j] \quad (4)$$

In Figure 8(a) we compare ICA-Tune with standard fine-tuning for increasing size of $D$. In both cases we finetune the attention parameters of all layers. To ensure comparability between the ICA-Tune and standard fine-tuning setup we generate a fixed dataset $D$ of $N$ of training examples for a task $\tau$. Both methods sample from $D$. We use batch size of 1 for ICA-Tune and 16 for standard fine-tuning. This choice ensures that both the setups train on the same number of examples per training step. We generate a separate set of

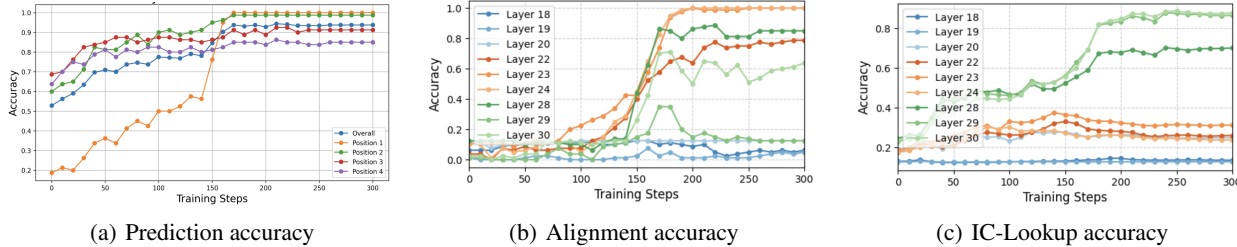

(a) Prediction accuracy  (b) Alignment accuracy  (c) IC-Lookup accuracy

*Figure 7.* Learning dynamics of ICA-Tune with increasing steps for Qwen2.5-3B model. (a) Prediction Accuracy for four different seeds: average and at different positions along the y-phrase. **Maximum jump for $y$ tokens at the start of a y-phrase.** (b) The gains are due to the emergence of $\mathbf{x}, \mathbf{y}$ alignment. The plot shows extracted alignment accuracy at different LLM layers. **Alignment accuracy high in middle layers (22,23) and abruptly rises between steps 120 and 170.** (c) IC-Lookup ability is seen in higher layers (28, 29) above the layers where alignment emerges. This shows that alignment is needed for formation of informative induction heads.**

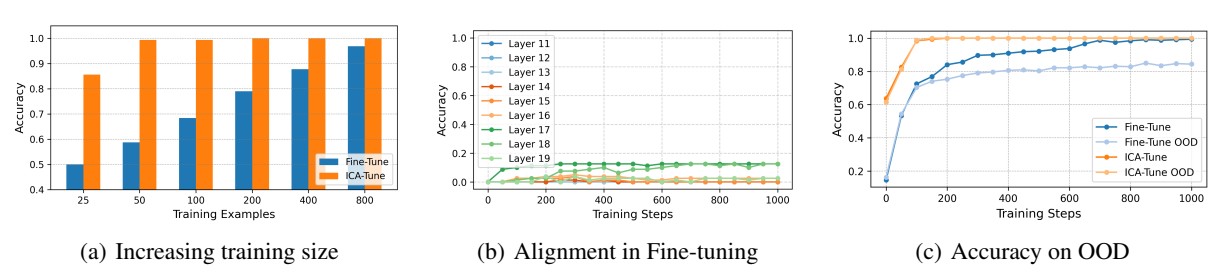

(a) Increasing training size  (b) Alignment in Fine-tuning  (c) Accuracy on OOD

*Figure 8.* Comparing ICA-Tune with standard fine-tuning. (a) Accuracy with different number of training examples. **ICA-Tune requires fewer training examples to achieve good accuracy.** (b) Alignment accuracy for fine-tuning. **Alignment does not emerge in any of the layers unlike in ICA-Tune** (Figure 5(b)). (c) Validation accuracy on IN and OOD sets of fine-tuning Vs ICA-Tune. **ICA-Tune convergence faster and generalizes better to out-of-distribution (OOD) validation examples compared to normal fine-tuning.**

$M = 10$ examples for validation. The validation set has individual examples for standard fine tuning and for ICA-Tune we prepend each validation example with $k$ examples from the training set. We make the following observation:

**ICA-Tune is significantly more sample efficient than standard fine-tuning.** ICA-Tune achieves almost 100% accuracy with just 50 labeled instances whereas standard fine-tuning does not reach that accuracy even with 800 examples. We explain this difference by observing the alignment accuracy of intermediate layers in the fine-tuned model.

**Standard fine-tuning does learn to align.** Comparing Figure 8(b) and Figure 5(b) that show alignment accuracy of the regular fine-tuned model and ICA-Tuned model respectively, we observe that fine-tuning does not uncover alignments unlike ICA-Tune. In ICA-Tune, the alignment causes the formation of the induction heads for in-context learning. Such shortcut is not available to regular fine-tuning.

**OOD Generalization of ICA-Tune.** ICA-Tune decomposes the Seq2Seq task into two key steps: (1) learning the alignment between input and output tokens, and (2) leveraging existing induction heads to perform associative lookups using the aligned keys. This decomposition en-

hances the model's capacity for out-of-distribution (OOD) generalization. In Figure 8(c), we compare the performance of ICA-Tune and standard fine-tuning on two distinct validation sets. For the first validation set, the input sequences $\mathbf{x}$ are sampled from the same CFG as the training set, ensuring an in-distribution evaluation. For the OOD validation set, $\mathbf{x}$ sequences are generated as random permutations of the $x$ vocabulary, deliberately designed to deviate from the training CFG. As shown in the figure, ICA-Tune does not distinguish between the two sets and converges faster to near perfect accuracy on both. In contrast, since fine-tuning did not learn alignments, it likely overfitted on the whole $\mathbf{x}$ sequences and shows poorer OOD generalization.

### 4.4. Alternative Alignment Functions

So far, our experiments were with monotonic alignments with $\lambda = \infty$ in Eq 2. Even for monotonic alignments, pre-trained Llama models fail to learn alignments in-context, and we introduced ICA-Tune that causes emergence of input-output alignment given already formed induction head. We next explore if ICA-Tune can also learn non-monotonic alignments arising out of small values of $\lambda$. For example, with $\lambda = 0$, the $p$th y-phrase may align with any of $m$ $\mathbf{x}$ tokens. All the training parameters and dataset values are

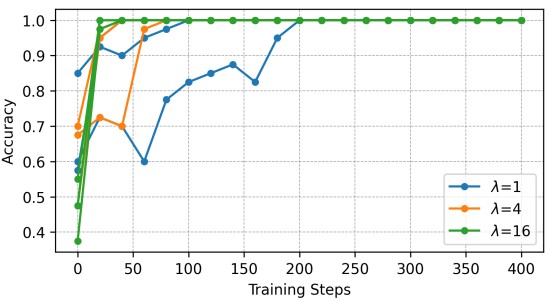

*Figure 9.* Training progress for varying $\lambda$s in the sampled alignment function $A_\tau(.)$ (see Equation 2). Larger $\lambda$ leads to monotonic alignment that is easier to learn than smaller $\lambda$s

same as previous experiments and $m = 8$, $c = 1$. Figure 9 shows accuracy as a function of fine-tuning steps for three different values of $\lambda$. We show runs with three seeds for each $\lambda$. For a large $\lambda = 16$, for all three seeds ICA-Tune learns fast. For $\lambda = 4$, one of the three seeds shows slower convergence. For $\lambda = 1$, convergence is the worst. For one of the seeds 200 steps were required as against just 25 for the monotonic case. We observe that for small values of $\lambda$, more training data is required to fine-tune the model.

## 5. Discussion: Can transformers learn Seq2Seq alignments in-context?

We saw that the pre-trained LLMs fail to in-context learn Seq2Seq alignments and required additional fine-tuning. Here, we ask if in-context learning of general non-monotonic alignments can be expressed at all in causal transformers. We informally conjecture that in-context learning of alignments for arbitrary $m$, $K$ values cannot be expressed in causal transformers with a fixed number of parameters. We present a discussion of why this conjecture might hold.

Consider the special case where the y-phrase length $c$ is 1 making the length of the input and output is the same $m = n$, and $A_\tau(j) \in \{1 \dots, m\}$, that is each token in **y** aligns to any arbitrary token position in **x**. $P_\tau(y_j | \mathbf{x}_{A_\tau(j)})$ and $A_\tau(j)$ has to be estimated in-context via induction heads from a set of $k$ examples $D = \{(\mathbf{x}^1, \mathbf{y}^1), \dots (\mathbf{x}^k, \mathbf{y}^k)\}$. This defines a maximum likelihood estimate of $A_\tau$ on the data $D$ as:

$$\max_{A_\tau} \sum_{i=2}^{k} \sum_{j=1}^{m} \log \underbrace{\frac{\sum_{r=1}^{i-1} \sum_{s=1}^{m} \delta([y_j^i, \mathbf{x}_{A_\tau(j)}^i] = [y_s^r, \mathbf{x}_{A_\tau(s)}^r])}{\sum_{r=1}^{i-1} \sum_{s=1}^{m} \delta(\mathbf{x}_{A_\tau(j)}^i = \mathbf{x}_{A_\tau(s)}^r)}}_{\text{MLE for } P_\tau(y_j^i | \mathbf{x}_{A_\tau(j)}^i) \text{ from previous examples}}$$

Informally, this objective implies that the estimated $A_\tau()$ should maximize the agreement in the mapped $x$ to $y$ vocabulary across the $mk$ occurrences. The standard solution is using the EM algorithm (we present the classical alignment learning EM algorithm in Appendix F). The algorithm alter-

nates between estimating the distribution over $x$ tokens that each $y$ token maps to, and estimating the alignment distribution. This would require memory of size $O(Vm + m^2)$ where $V$ is the number of distinct $y$ tokens in the in-context examples. Since consensus across all $k$ examples is needed, maintaining this memory distributed across the $2km$ states of the transformer is not possible for a causal model. Also, expressing the EM update in terms of transformer operations seems non-trivial without additional chain-of-thought or scratch memory.

## 6. Conclusions

In this paper we presented the first ever formal evaluation of the in-context learning abilities of pre-trained LLMs on structured sequence to sequence prediction tasks. To allow for a systematic exploration without interference from the LLMs training datasets, we design a realistic synthetic generator of a structured language pair. Our study shows that LLMs fail to in-context learn Seq2Seq tasks even when they succeed in simpler corner cases of scalar key-value mappings and language completion. Via counterfactual experiments using pre-aligned sequences, we attribute the failure to the inability of the LLM to figure out new x-y alignments in-context. We propose ICA-Tune to fine-tune the LLM with in-context examples. ICA-Tune harnesses existing induction circuits to infer the token distributions in-context while fine-tuning attention parameters to learn the alignments. Via mechanistic probes we show the emergence of the alignment capability in the middle layers of the LLM that lead to the formation of informative keys to induction heads in higher layers — all this with just a next-token prediction loss. In contrast, standard fine-tuning fails to learn alignments, and that leads to inefficient learning and poor OOD generalization indicative of the model's failure to uncover the decompositional structure of the task.

This work brings our several avenues of future research: formally analyzing the representation power of transformers for in-context learning alignments, designing pre-training datasets that can cause the emergence of ICL over structured Seq2Seq tasks following a strong prior e.g., monotonicity, and transferring these discoveries to real datasets.

**Acknowledgment** We thank SBI Foundation's Data Analytics Hub at IIT Bombay and Google Research India for supporting this research.

## Impact Statement

This paper presents work whose goal is to advance the field of Machine Learning. There are many potential societal consequences of our work, none which we feel must be specifically highlighted here.

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

# A. More detailed related work

Many studies resort to testing on synthetically generated data in order systematically analyze the reasons behind the empirical performance of ICL, without leakage from the vast pre-training set of large models. Theoretical and mechanistic interpretation on such synthetic settings have been used to provide various explanations for how a transformer model implements ICL. Using synthetic data and models trained from scratch for regression tasks, many studies claim that the transformer's self attention across layers implements the gradient descent algorithm (Garg et al., 2022; Ahn et al., 2023; Akyürek et al., 2023; Panwar et al., 2024; Mahankali et al., 2024; Zhang et al., 2023a; Von Oswald et al., 2023; Ahn et al., 2023; Zhang et al., 2023b; Giannou et al., 2024; Yang et al., 2024; Gatmiry et al., 2024). However, these conclusions have been found not to hold for discrete NLP tasks which is our focus (Deutch et al., 2024; Li et al., 2024; Shen et al., 2024). Another hypothesis is that ICL performs Bayesian task selection (Reynolds & McDonell, 2021; Min et al., 2022; Pan et al., 2023; Shi et al., 2024). However, this hypothesis does not explain the capability of LLMs to learn new input-output mappings or new languages. (Abernethy et al., 2024) present a mechanism by which a pre-trained transformer can in-context learn sparse retrieval tasks, including learning the task-specific delimiter between example and label, and between successive examples. Their approach is to show that their exist transformer parameters that can be oriented to online optimize for the best delimiter under prior biases over a small set of delimiters. They require separate heads for each possible delimiter pair. Also, for discovering coefficients of the sparse retrieval task they also show existence of transformer parameters that can online learn the coefficients.

# B. More details of the structured sequence to sequence model (Section 3.1)

## B.1. Generating Input Sequence x Using a Context-Free Grammar

To generate an input sequence $\mathbf{x} = x_1, x_2, \ldots, x_m$ consisting of $m$ discrete tokens, we employ a probabilistic context-free grammar (CFG). This allows us to systematically produce structured sequences through a series of production rules. We chose a simple two-level CFG as follows. First we partition the vocabulary of $x$ tokens $\Sigma_\tau$ into two disjoint parts of almost equal size $U_1$ and $U_2$, and use these to define the grammar using non-terminals $R, X, Y$ as

$$R \to XY | YX$$
$$X \to \text{all permutations of } U_1$$
$$Y \to \text{all permutations of } U_2$$

- **Root Symbol** ($R$): The start symbol $R$ generates a concatenation of two sub-sequences, $X$ and $Y$, in either order ($XY$ or $YX$).
- **Sub-sequence** $X$: The sub-sequence $X$ consists of all possible permutations of the set $U_1$.
- **Sub-sequence** $Y$: Similarly, the sub-sequence $Y$ includes all permutations of the set $U_2$.

Each transition in the grammar is assigned an equal probability, ensuring that all valid expansions are equally likely to occur.

An example PCFG for $m = 3, \Sigma_\tau = \{A, B, C\}$ is

$$R \to XY | YX$$
$$X \to AB | BA$$
$$Y \to C$$

## B.2. Model to sample a y-phrase given an input $x$ token

We model the generation of a y-phrase of a given length, say $n$, using a probabilistic finite state automaton.

**Background on Probabilistic Finite State Automaton (PFA)** A PFA is defined by an alphabet $\Sigma$, a finite set of states $S$, an initial state $s_0$, and a state transition distribution $T : S \times \Sigma \times S \to [0, 1]$ where $\sum_{x, S'} T(x, S'|S) = 1 \, \forall S$[3]. Accordingly,

---

[3]In general, a PFA might also attach a distribution over initial and accepting states, but we did not need that flexibility for this generation task.

**Layer 3: Copy over label based on similarity of context**

| A | C | : | c | i | ; | B | E | : | f | p | ; | E | A | : | p | c | ; | A | B | : |
|---|---|---|---|---|---|---|---|---|---|---|---|---|---|---|---|---|---|---|---|---|
|   |   |   | A | C |   |   |   |   | B | E |   |   |   |   | E | A |   |   |   | 11 |

**Layer 2: Copy aligned x token as key using position similarity**

| A | C | : | c | i | ; | B | E | : | f | p | ; | E | A | : | p | c | ; | A | B | : |
|---|---|---|---|---|---|---|---|---|---|---|---|---|---|---|---|---|---|---|---|---|
| 1 | 2 |   | 1 | 2 |   | 1 | 2 |   | 1 | 2 |   | 1 | 2 |   | 1 | 2 |   | 1 | 2 |   |

**Layer 1: Context formed by assigning relative position ids to each x and y token**

| A | C | : | c | i | ; | B | E | : | f | p | ; | E | A | : | p | c | ; | A | B | : |
|---|---|---|---|---|---|---|---|---|---|---|---|---|---|---|---|---|---|---|---|---|

*Figure 10.* A possible three layer induction circuit to support ICL on our Seq2Seq task with monotonic alignment. Here $k = 2, m = 2, c = 1$. The first layer that assigns relative position ids to the x and y tokens needs to exploit strong priors about delimiters and example separators (Abernethy et al., 2024). The second layer that aligns based on matching position works only for monotonic alignments. The last layer is the standard induction head.

the probability of generating a sequence $\mathbf{x}$ is given as $p_{\text{PFA}}(\mathbf{x}) = \sum_{s_0,\dots,s_n} \prod_{i=1}^{n} T(s_{i-1}, x_i, s_i)$. To generate a sequence of length $n$ from a PFA we start with $S_0$. For each $i$ from 1 to $n$, we sample a $x_i, S_i$ from $T(x, S|S_{i-1})$. The output sequence $\mathbf{x} = x_1, \dots, x_n$ is the sequence of output symbols. Without loss of generality we will assume that $S_0 = 0$ is always the start state and will denote PFA as PFA$(\Sigma, S, T)$.

**Method of constructing Probabilistic Finite State Automaton (PFAs)** We present how we sample the vocabulary $\Sigma$, set of states $S$, and the transition probabilities $T(x, S'|S)$

(1) $\Sigma$: Sample an alphabet size $V$ uniformly from the interval $(V_{\min} = 4, V_{\max} = 18)$. Sample a language-specific alphabet $\Sigma$, containing $V$ symbols, uniformly (without replacement) from a shared symbol set $W$ (with $|W| = c_{\max} = 26$).

(2) $S$: Sample a number of states $s$ uniformly from the interval $(s_{\min} = 4, s_{\max} = 12)$. Given this value, define a set of automaton states $S = \{S_1, \dots, S_s\} \cup \{S_0\}$. Without loss of generality we will assume that $S_0 = 0$ is always the start state.

(3) $T(x, S'|S)$: For each $S_i$, choose a number of outgoing edges $m_i$ uniformly from $(m_{min} = 1, m_{max} = 4)$. Then, construct a set of edges $(S_i, z_j, S_j)$, where all $z_j$ are sampled uniformly without replacement from $\Sigma$, and all $S_j$ are sampled uniformly without replacement from $\{S_1, S_2, \dots S_n\}$. Assign a probability of $T(z_j, S_j|S_i) = \frac{1}{m_i}$. Set $T(z', S'|S_i) = 0$ for all other pairs $(z', S')$ that do not correspond to the generated edges.

## C. Transformer Architecture

We assume a pre-trained decoder-only transformer model which have been shown to be capable of ICL. For new tasks, ICL has been attributed to the formation of induction heads. We present a brief overview of how causal self-attention computations across multiple layers could enable this capability. The input to the transformer is sequence $X : X_1, \dots X_N$ of $N$ discrete token ids from a vocabulary of $V$ tokens, which after mapping via a $d$-dimensional embedding matrix $E \in \mathbf{R}^{V \times d}$ generates $Z \in \mathbf{R}^{d \times N}$. The transformer comprises of $L$ layers, and $H$ heads per layer. The computation at each $\ell$ given input $Z$ can be expressed as a function

$T_\ell : \mathbf{R}^{d \times N} \mapsto \mathbf{R}^{d \times N}$ as follows:

Parameters $\theta_\ell = \{W_\ell, P_\ell, (V_{\ell,h}, Q_{\ell,h} K_{\ell,h}) : h \in [H]\}$

$$A_h \leftarrow \text{softmax} \circ \text{mask}(Z^T Q_{\ell,h}^T K_{\ell,h} Z) \in \mathbf{R}^{N \times N} \quad \forall h \in [H]$$

$$Z \leftarrow Z + \text{FF}_{W_\ell}(P_\ell \, \text{concat}(V_{\ell,h} Z A_h : h \in [H])) \in \mathbf{R}^{d \times N}$$

Here mask ensures that every position $i$ only attends to positions before it, and is called a causal mask. After the last layer $T_L$, the output $Z$ is multiplied by the embedding matrix and then softmax-ed to get a probability distribution over tokens.

This makes the overall transformer implement a function of the form

$$T_\theta(X) = \text{softmax}(E.T_L \circ \ldots \circ T_1(E^T X + \mathbf{p})) \tag{5}$$

where $\theta = \cup_{\ell=1}^N \{W_\ell, P_\ell, (V_{\ell,h}, Q_{\ell,h}K_{\ell,h}) : h \in [H]\} \cup E$ denotes all the parameters of the transformer, $\mathbf{p}$ denotes a vector of position embeddings.

## D. Experiment setup

For fine-tuning, we employ Low-Rank Adaptation (LoRA), which efficiently adapts pre-trained language models by injecting trainable low-rank updates into specific model parameters.

**LoRA Hyperparameters**   We use the following LoRA configuration for the experiments:

- **LoRA Rank (LoRA$_R$)**: Set to 16. This rank determines the dimensionality of the low-rank decomposition applied to the weight matrices.
- **Scaling Factor (LoRA$_\alpha$)**: Set to 8. This hyperparameter controls the scaling of the low-rank updates during training to ensure stability and effective learning.
- **Dropout (LORA_DROPOUT)**: Set to 0.05 to introduce regularization and prevent overfitting in the low-rank layers.

**Training Configuration**   We fine-tune all attention parameters, specifically the $Q$, $K$, and $V$ matrices of the transformer. We use a learning rate $= 2e^{-4}$ for training. We use the Adam optimizer along with a linear decay learning rate scheduler.

## E. Additional Experiments

### E.1. Fine-tuning only a single layer

For this task, and particularly with monotonic alignments, ICA-Tuning even a single layer of the Llama model suffices to cause the emergence of input-output alignment. However, some layers are significantly more effective than others when fine-tuned. In Figure 11(a) we present the accuracy with each layer fine-tuned. This graph shows that fine-tuning the attention parameters of one of the middle layers $\ell = 13$ is most effective. Fine-tuning attention of higher layers (denoted by lighter shade) has very effect on enhancing ICL accuracy. Early layers (less than 5) helps more than higher layers but still less than the middle layer 13. Subsequently we study the alignment and IC-Lookup accuracy, and the attention heatmaps as layer 13's attention parameters are fine-tuned. The emergence of alignment and IC-Lookup accuracy are similar as in the case when all attention parameters were fine-tuned.

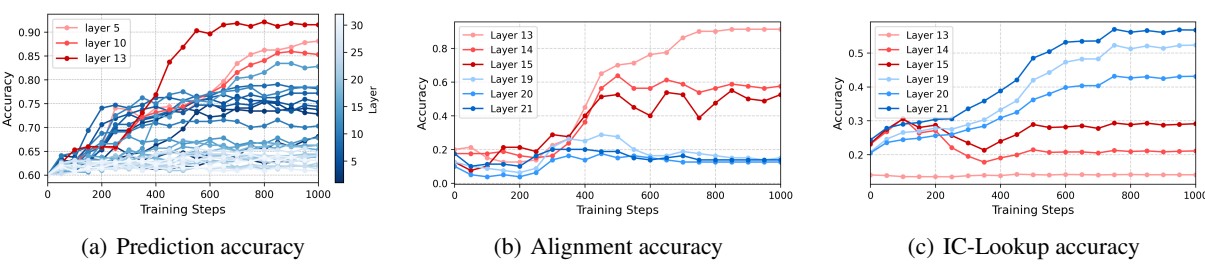

|     (a) Prediction accuracy     |     (b) Alignment accuracy     |     (c) IC-Lookup accuracy     |

*Figure 11.* Learning dynamics of ICA-Tune with increasing steps. (a) Prediction Accuracy when fine-tuning different attention layers. Higher layers are lighter shades than lower layers. **Maximum gains is from a middle layer $\ell = 13$.** (b) Emergence of alignment accuracy at different LLM layers when layer $\ell = 13$ is fine-tuned. **Alignment accuracy high in middle layers and emerges around steps 430.** (c) Emergence of IC-Lookup accuracy with fine-tuning over LLM layers when layer $\ell = 13$ is fine-tuned. **IC-Lookup ability emerges in higher layers around steps 470**

### E.2. Controlling for Prompt Length

To ensure that the improved performance with pre-aligned prompts is not merely due to having more tokens in the input, we repeat the experiment with a reduced number of examples (14 instead of 16) in the pre-aligned sequence. This adjustment brings the total number of tokens in the pre-aligned prompt roughly in line with that of the standard prompt.

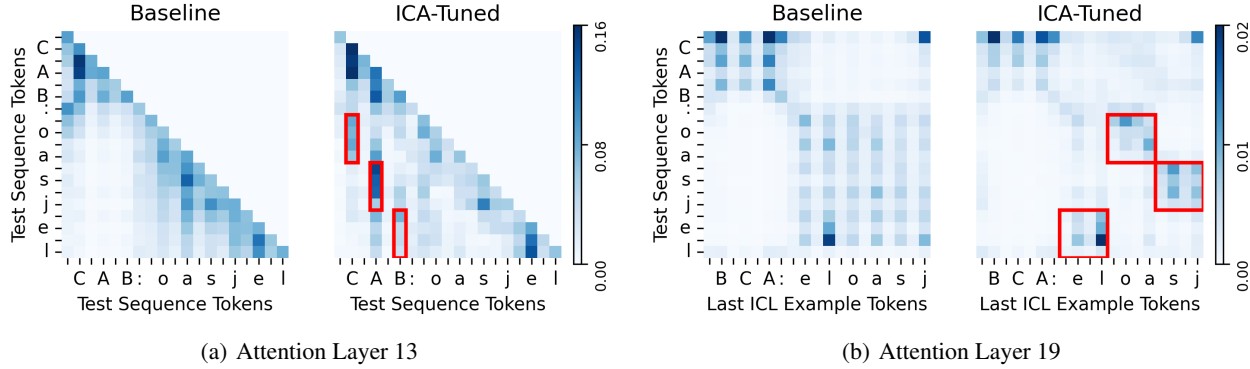

(a) Attention Layer 13          (b) Attention Layer 19

*Figure 12.* (a) Attention heatmaps for layer 13 of the baseline and fine-tuned model. The heatmaps show how attention is distributed between the y-tokens (target tokens) and the corresponding x-tokens (input tokens). In the fine-tuned model, initial y-tokens tend to focus on their corresponding x-tokens more consistently compared to the baseline. (b) The attention heatmaps for layer 19 between tokens from the test example (on the y-axis) and to tokens of nearest IC example (on the x-axis).

Table 3 reports the accuracy for all three models under the standard, full pre-aligned, and reduced pre-aligned settings. Even after equalizing the prompt length, the reduced pre-aligned setup continues to significantly outperform standard prompting. This reinforces the importance of alignment rather than just prompt size in driving ICL performance.

| Model | Standard | Pre-Aligned | Reduced Pre-Aligned |
|-------|----------|-------------|---------------------|
| Llama-3.2-1B | 27.50 | 100.00 | 99.38 |
| Llama-3.2-3B | 30.63 | 95.94 | 95.00 |
| Llama-3.2-8B | 36.25 | 98.44 | 99.38 |

*Table 3.* Prediction accuracy (%) under different prompt formats for each model. Reducing the number of examples in the pre-aligned prompt still leads to higher accuracy than standard prompting, suggesting that alignment—not token count—is the key factor.

## F. EM algorithm for learning alignments

As an exercise, let us go over a standard method of discovering alignment following the EM algorithm (Dyer et al., 2013). Let $a_{ji}$ denote the alignment probability that $A_\tau(j) = i$, and $\beta_{uv} = P_\tau(y_j = u | x_i = v)$ denote the soft $x - y$ mapping dictionary. The data likelihood

$$\sum_{i=1}^{k}\sum_{j=1}^{m}\log\sum_{p} P(A_\tau(j) = p)P_\tau(y_j^i | x_p^i) = \sum_{i=1}^{k}\sum_{j=1}^{m}\log\sum_{p} a_{jp}\beta_{y_j^i x_p^i} \tag{6}$$

Maximizing the above objective w.r.t parameters $a_{jp}, \beta_{uv}$ can be achieved via the EM algorithm, and the iterative update equations of the parameters can be shown to be

$$a_{jp} = \frac{\sum_{i=1}^{k} a_{jp}\beta_{y_j^i x_p^i}}{\sum_{i=1}^{k}\sum_{q} a_{jq}\beta_{y_j^i x_q^i}} \tag{7}$$

$$\beta_{uv} = \frac{\sum_i \sum_{j:y_j^i=u} \sum_{p:x_p^i=v} a_{jp}\beta_{uv}}{\sum_i \sum_j \sum_{p:x_p^i=v} a_{jp}\beta_{y_j^i v}} \tag{8}$$

There are several challenges to implementing these iterations in the decoder-only transformer, one option is to assume that the hidden state $Z$ of the last input time-step maintains the $\beta_{uv}$ and $a_{ji}$ values. This would require the hidden vector to be of size $O(Vm + m^2)$ where $V$ is the number of distinct $y$ tokens in the in-context examples. Also, expressing the update equations 7 in terms of transformer operations is not obvious. Based on these, we conjecture that in-context learning of alignments over unseen tasks may not be feasible in a transformer with a fixed small embedding size.

