# OpenReview forum: "The Missing Alignment Link of In-context Learning on Sequences"
_ICML.cc/2025/Conference — ICML 2025 poster_

### Official Review · Reviewer_kbAw · 2025-03-04

**Overall Recommendation:** 3

**Summary:**

Authors study the limits of LLMs’ abilities for in-context learning, focusing on learning sequence to sequence alignment (in the machine translation sense of the word). Authors design synthetic experiments that probe said ability and demonstrate that several modern llama 3 variants do indeed fail to learn alignment in-context. To combat this, authors introduce ICA-Tune, a PEFT strategy that learns to adjust the attention weights within the LLM. Authors analyze the fine-tuned models and demonstrate that ICA-Tune learns input-output alignment by forming new induction circuits in the middle layers of the model.

**Claims And Evidence:**

The main claims in the paper are, from my perspective:
1. (L95) "the inability of modern LLMs to in-context learn structured Seq2Seq tasks" - stemming from their inability to learn the task-specific alignments (in MT sense) beyond very short sequences;
2. That the proposed method, ICA-Tune, can alleviate said problem through a mixture of parameter-efficient fine-tuning and in-context learning;
3. The existence of several phenomena within LLM representations, e.g. the specific induction circuit reported in (L324).

To support these claims, authors design a synthetic in-context-learning problem generator inspired by classical machine translation and demonstrate (with reasonable ablation) that the LLM in question fails to learn the alignment. They then demonstrate how this can be rectified with ICA-Tune. In my opinion, claims 2 and 3 are properly supported, but the first claim is arguably too general.


**Overclaim about generality.** My main concern is that the paper oft makes general claims about “modern LLMs” (e.g. L95-97 “the inability of modern LLMs to …”) that are, in my view, not properly supported and may mislead readers. Authors only consider Llama 3 8B in the main paper and Llama3.2-1B & 3B in appendix E.1: all  relatively small and from the same (albeit popular) model family. This leaves several potential ways that the hypothesis may turn out false:

**A. Emergent abilities?** it could be that modern LLMs do learn alignment, but it only emerges only after a certain model size (e.g. 70B, 405B). LLMs have been known to ‘sprout’ new emerging abilities in the past [1]. From the current analysis, it is unclear if modern LLMs are fundamentally unable to learn alignment (as the authors claim) or if this is only a failing of the particular LLMs chosen for the study.

**B. Model idiosyncrasy?** while less likely than the previous hypothesis, it is possible that there is something about specific Llama 3 training or inference conditions that affect their ability to learn alignment. This can be eliminated by testing on other models: latest qwen 2.5 / mistral / deepseek / qwq model variants.

Unless authors can eliminate these possibilities, I would argue that the claims need to be rewritten substantially (e.g. the inability of modern LLMs -> the apparent inability of such and such LLM types under such conditions) or toned down.

Note that double-checking this claim does not require significant expenditure of resources: to the best of my knowledge, one may test the state-of-the-art LLMs’ ability to learn alignment **using public APIs**, without any specialized hardware. This includes both open models on free tier endpoints (e.g. lmarena.ai, deepseek R1) and commercial models (e.g. openai API, anthropic, google, etc), since the test only requires in-context learning and inference. . If authors find that a diverse set of SoTA models consistently fails to learn alignment, it would strengthen the claims significantly.

**Essential References Not Discussed:**

Authors investigate a narrow (but important) capability of modern LLMs and reviews related works in S2, and, to the best of my knowledge, their review is sound (but not encyclopedic). Though, it is not impossible that I missed some other relevant work.

**Experimental Designs Or Analyses:**

While the experiments are limited to synthetic problems, they are generally sound. Authors consider reasonable setups and perform additional ablations (e.g. p.4 right) to verify their observations. My main concern about the choice of LLMs and the general claims about LLM abilities is described above in the "Claims And Evidence" section.

**Methods And Evaluation Criteria:**

Overall, they do indeed make sense. The paper makes a deliberate choice to evaluate on synthetic data, which determines both its strengths and weaknesses. On the positive side, the controlled experiment allows authors to better isolate the phenomenon, vary experiment parameters and design counterfactual experiments for ablation. On the negative, it leaves a chance that the proposed solution (ICA-Tune) may not transfer as well to real world data.

**Other Comments Or Suggestions:**

**Minor suggestions**

- please close opened files (e.g. dataset.py L27-31)
- model loading (icatune.py L39, 44) makes it unclear which llama3 are you loading and what format must one prepare this model in
- please check the code for unused dependencies (e.g. re in icatune.py)

**Other Strengths And Weaknesses:**

**Strengths**
A simple and practical patch for a specific problem, no overthinking. Even if it doesn’t generalize to all LLMs, it’s still useful to many.
Within the one synthetic task they consider, author go to commendable length to ablate their findings (S3, S4). Analysis of the ICA-Tune-d models, if mostly mechanistic, is a great addition.

**Weaknesses**
As I specified earlier in "Claims And Evidence", I believe the main weakness of the paper to be overclaim: authors make conclusions about "modern LLMs" in general, but only consider one model family, and only relatively smaller models.

While the paper could also be strengthened by real world experiments to confirm the efficacy of ICA-Tune, it appears to be a deliberate choice and not a weakness.

**Questions For Authors:**

(minor, extension of OOD generalization experiments) From a practitioner's point of view, are there feasible ways to extend ICA-Tune to 'pre-patch'  known LLMS ahead of time so they would be able to learn alignment in-context for a broader range of unknown ICL tasks?

**Relation To Broader Scientific Literature:**

The paper relies on popular prior work in the LLM community: the chosen model family, LoRA adapters, circuits, etc.

**Theoretical Claims:**

To the best of my abilities, the main claims in the paper are empirical, not theoretical.

---

> ### Author Rebuttal · Authors · 2025-04-01
>
> Thank you for the thoughtful feedback. We will address the issues with the supplementary material in the final submission. Below, we respond to the other concerns.
>
> A. We repeated the experiments of Figure 3 on multiple LLMs. We observe similar trends. We report below the numbers for m = 8, c = 1 where we compare the Standard and Pre-Aligned prompting method. We observe that even large models exhibit poor sequence prediction accuracy. Also, interleaving the x and y tokens as a pre-aligned token sequence leads to significant jump in accuracy, indicating that lack of alignment is a major reason for poor performance.
> | Model            | Standard | Pre-Aligned |
> |-----------------|----------|-------------|
> | Llama-3.3-70B-Instruct-Turbo   |     20     |      91.25       |
> | Llama-3.1-405B-Instruct-Turbo   |     31.25     |      98.75       |
> | gpt-4o   |     36.25     |       100.0     |
> | claude-3-7-sonnet-20250219   |     51.25     |       82.5      |
>
> B. We also repeated the experiments of Figure 4 on the learning dynamics of ICATune on a Qwen2.5-3B model.  The results can be found in this [link](https://drive.google.com/file/d/1R90cBSwCaBkumSwibopWckyVIEDZxpaj/view?usp=sharing).  We observe the same conclusions over the three plots --- (1) Sudden jump in prediction accuracy of the y tokens at the start of a y-phrase, (2) Emergence of alignment in middle layers (23,24) of the LLM. (3) Followed by IC-Lookup ability in higher layers (28,29) above the layers where alignment emerges.  This Qwen model is smaller than the Llama model-3 and yet we observe the same pattern.
>
> C. We also present results on real translation dataset for three language pairs. Please see the results in the response to reviewer roK1.

---

### Official Review · Reviewer_D59M · 2025-03-12

**Overall Recommendation:** 4

**Summary:**

This paper systematically investigates the in-context learning (ICL) capabilities of large language models (LLMs) on sequence-to-sequence (Seq2Seq) tasks. The analysis reveals that LLMs struggle to align input and output sequences for longer inputs, limiting their ICL effectiveness. To address this, the authors propose ICA-Tune, a method that fine-tunes attention parameters using in-context examples and a next-token prediction objective. Compared to standard fine-tuning, ICA-Tune demonstrates superior sample efficiency and better generalization to out-of-distribution (OOD) instances.

**Claims And Evidence:**

see strengths and weaknesses

**Essential References Not Discussed:**

see strengths and weaknesses

**Experimental Designs Or Analyses:**

see strengths and weaknesses

**Methods And Evaluation Criteria:**

see strengths and weaknesses

**Other Comments Or Suggestions:**

None

**Other Strengths And Weaknesses:**

Strengths
1. This paper presents the first formal evaluation of ICL on Seq2Seq tasks, which has significant practical implications for applications such as instant translation and text-to-SQL.
2. The counterfactual experiments provide valuable insights, highlighting that alignment is the missing link in Seq2Seq ICL. The mechanistic evaluation of alignment emergence in middle layers is particularly insightful.
3. The proposed ICA-Tune method demonstrates clear improvements in sample efficiency and OOD generalization, supported by rigorous empirical experiments.

Weaknesses
1. The analysis is primarily conducted on synthetic data, justified by the need to avoid data contamination (L33). However, it would be beneficial to evaluate ICA-Tune on tasks that LLMs have encountered during pre-training, such as translation. This could help determine whether Seq2Seq ICL is an online learning process (as assumed in this paper) or a task-level generalization mechanism that composes pre-existing knowledge [1]. I'd like to hear the authors' opinion about the other ICL perspective (generalization via pre-existing knowledge composition [1]).
  - [1]  What Do Language Models Learn in Context? The Structured Task Hypothesis, ACL'24
2. After fine-tuning with ICA-Tune, does the LLM’s performance degrade on general Seq2Seq tasks (e.g., translation)? If so, this could indicate a limitation of ICA-Tune’s applicability. Is it possible to explore methods that specifically learn alignment in a way that generalizes across diverse Seq2Seq tasks?
3. In L84, references to previous studies would be more effective if explicitly cited.

**Questions For Authors:**

None

**Relation To Broader Scientific Literature:**

see strengths and weaknesses

**Theoretical Claims:**

see strengths and weaknesses

---

> ### Author Rebuttal · Authors · 2025-04-01
>
> Thank you for the constructive suggestions. We address some of the concerns below.
>
> (1) Pre-existing knowledge composition is a plausible hypothesis to explain ICL on real tasks. However, the experiments in [1] were on scalar prediction tasks.  We conjecture that for structured sequence prediction tasks, the alignment of output substructures with relevant input phrases is necessary for compositional generalization, irrespective of whether ICL is due to online learning or knowledge composition, or a combination of the two.
>
> (2) Catastrophic forgetting of previous tasks, for example translation of real tasks after fine-tuning on synthetic tasks, is present in both standard fine-tuning and ICA-Tune. However, on variants of the same synthetic task, ICA-Tune does generalize better as we discuss in the last paragraph of Section 4.2.  We have been thinking hard about methods that specifically learn alignment in a way that generalizes across diverse Seq2Seq tasks.  Section 5 presents a discussion on why such a capability could be difficult in standard causal transformers.  However, it may be possible to design multi-task in-context datasets for structured prediction tasks that enables LLMs to in-context learn alignments for similar tasks.
>
> (3) Thanks for pointing out. We will explicitly cite previous studies on L84 in the final version of the paper.

---

### Official Review · Reviewer_bbEZ · 2025-03-16

**Overall Recommendation:** 3

**Summary:**

This work presents an interesting case study on LLM's in-context ability on translation-style seq2seq problems. More specifically, this paper studies seq2seq problem involving both learning the alignment and the target-side vocabulary. This work creates synthetic tasks for the analysis to avoid training data leakage. The synthetic tasks are composed of a PCFG generating the source, PFA generating the target vocabulary, and a sampling function generating the alignment. By manipulating hyperparameters of the synthetic tasks and checking the attention weights, this work notices that LLM struggles at learning the alignment. Then, this work also proposes a fine-tuning-based method to improve this specific ICL ability and show that it is more effective than IID supervised learning.

**Claims And Evidence:**

This work mainly makes claims on two parts: (1) an analysis showing LLMs struggle at learning alignment through ICL; (2) a new fine-tuning method called ICA-Tune that helps LLMs to learn alignment.

I find the synthetic task-based analysis quite convincing. The experiment setup is clear and rigorous, and the attention weight-based analysis shows clear patterns.

I also believe that ICA-Tune improves the model's ability to learn alignment, but the comparison between ICA-Tune and standard SFT is a bit weird. Essentially, they work on two different tasks. The ICA-Tune is learning an in-context learning task, while the standard SFT is learning an IID seq2seq task. I also couldn't find related results on "latest PEFT methods" in the paper, as mentioned in line 108.

**Essential References Not Discussed:**

No essential missing references.

**Experimental Designs Or Analyses:**

See above.

**Methods And Evaluation Criteria:**

See above.

**Other Comments Or Suggestions:**

Line 384: does -> does not?
Figure 2 and Table 1 use very similar examples, but Figure 2 says "attention is non-monotonic" and Table 1 says "alignment is monotonic". This is really confusing, even though I can understand how the grammar works from the main text.

**Other Strengths And Weaknesses:**

n/a

**Questions For Authors:**

1. For ICA-Tune, do you only train on the k+1-th example for the k-shot learning input, or do you also train on all previous k outputs?

2. My understanding is that due to the use of PFA to generate y-phrases, each output is non-deterministic. Is my understanding correct? If so, when computing prediction accuracies, do you need to consider all correct outputs? And have you done experiments checking if this non-determinism makes learning alignment much harder?

**Relation To Broader Scientific Literature:**

This work can be seen as a good next step related to previous work studying LLM's ability to learn regular language. The synthetic experiment design is clean and hopefully will inspire more future work on detailed analysis of the limitations of LLM's in-context learning abilities.

**Theoretical Claims:**

No major theoretical contributions.

---

> ### Author Rebuttal · Authors · 2025-04-01
>
> Thank you for your thoughtful feedback. We respond to the questions below.
>
> (1) We train on all y-tokens, including the outputs from all previous k examples. For the comparison between standard fine-tuning and ICA-Tune, we scale the batch size for standard fine-tuning to match the number of examples used in each batch for ICA-Tune.
>
> (2) Yes, your understanding is correct. When computing accuracy, we evaluate predictions against all valid outputs defined by the PFA.  Yes, the learning could be harder, but we wanted to be close to real tasks where such non-determinism is expected.

---

### Official Review · Reviewer_roK1 · 2025-03-18

**Overall Recommendation:** 2

**Summary:**

This paper investigates a critical challenge in in-context learning for sequence-to-sequence tasks, where they find modern LLMs struggle to learn alignments between input and output sequences in-context. The authors first show that providing explicitly aligned in-context examples dramatically improves performance. They then introduce ICA-Tune, a fine-tuning method using in-context examples, which not only improves accuracy compared to standard supervised fine-tuning but also yields better sample efficiency and out-of-distribution generalization. A detailed mechanistic analysis demonstrates that ICA-Tune enables the emergence of input–output alignments in the middle layers of the LLM, even without direct supervision, thereby facilitating the formation of induction heads for effective token distribution learning.

**Claims And Evidence:**

The experimental results generally support the paper’s claims. However, several aspects could be further controlled or clarified:

1. When using pre-aligned prompting, the increase in prompt length (and thus the number of tokens) may provide the model with additional context and compute used in attention, potentially biasing the comparison.
2. Similarly, the in-context fine-tuning naturally benefits from extended sequences, raising the possibility that performance gains might partly stem from this increased seq len and computation rather than from the alignment learning per se.

**Essential References Not Discussed:**

no

**Experimental Designs Or Analyses:**

see other points

**Methods And Evaluation Criteria:**

see other points

**Other Comments Or Suggestions:**

please see other points

**Other Strengths And Weaknesses:**

Strengths:

1. The paper offers a clear and formal formulation of the seq2seq in-context learning problem.
2. The proposed ICA-Tune method is both simple and effective, demonstrating substantial improvements in performance.
3. The mechanistic analysis is thorough, with well-designed probes that provide insights into how alignment emerges within the model.

Weaknesses:

1. The notation in Section 3.1 is somewhat confusing. For example, in Equation 2, it is not immediately clear to me that q indexes a token in the input x while p serves as the phrase index. Clarifying that the task maps a token in x to an entire phrase in y would improve readability.
2. The task formulation is highly synthetic—each input token is aligned with a single output phrase. This simplification may not fully capture the complexity of real-world seq2seq tasks.

3, The improvement observed with pre-aligned in-context sequences might be partly due to the longer prompt (i.e., more tokens providing additional context and compute) rather than solely due to better alignment.

4, The comparison between ICA-Tune and standard fine-tuning might be influenced by the fact that the test examples in the ICA-Tune setting are less out-of-distribution compared to those in the standard fine-tuning setup.

Overall, while the study is valuable for understanding seq2seq ICL, the highly controlled synthetic nature of the experiments may limit the generality of the findings and robustness of their claims.

**Questions For Authors:**

please see other points

**Relation To Broader Scientific Literature:**

this paper is related to in-context learning mechanisms, particularly building on work about induction heads and this paper is extending this to the seq2seq domain.

**Theoretical Claims:**

there is no theoretical claims in the main paper, the paper is largely empirical and mechanistic

---

> ### Author Rebuttal · Authors · 2025-04-01
>
> Thank you for the detailed comments. We address the concerns below.
>
> (1) Thanks for pointing this out; we will make the clarification in the final version of the paper.
>
> (2) We agree the task is synthetic but it is related to the model followed in early statistical machine translation models. Further, we present an evaluation of ICA-Tune on real translation tasks described below. Here also, we observe that ICA-Tune is more sample efficient and generalizes better to OOD instances.
>
> We conduct standard fine-tuning with ICA-Tune on three real Machine Translation tasks — from English to {Lithuanian(lt), Hindi(hi), Tamil(ta)}. In each case, we evaluate both on the in-distribution (ID) test set and on an out-of-distribution (OOD) test set. Details of the experiments and the results appear below.
>
> ### Experiment Details
>
> | Model Used     | Llama-2 7B                                    |
> |----------------|-----------------------------------------------|
> | Number of Training Examples | 40,000                                 |
> | Train Dataset  | [Europarl (For En-Lt)](https://opus.nlpl.eu/Europarl/corpus/version/Europarl), [Samanantar + BPCC + Speech Lab (For En-Hi)](https://huggingface.co/datasets/hamees/INDiC-BPCC-hq), [Samanantar (For En-Ta)](https://huggingface.co/datasets/ai4bharat/samanantar) |
> | Test Dataset   | [Flores (In-Domain for all three language pairs)](https://huggingface.co/datasets/openlanguagedata/flores_plus), [Tanzil (Out-of-Domain for En-Hi, En-Ta)](https://opus.nlpl.eu/Tanzil/corpus/version/Tanzil), [EMEA (Out-of-Domain for En-Lt)](https://opus.nlpl.eu/EMEA/corpus/version/EMEA) |
> | Epochs          | 2                                             |
> | Batch Size     | 2                                             |
> | Learning Rate  | 0.0002                                        |
> | LR Scheduler   | Linear                                        |
> | Warmup Steps   | 500                                           |
> | Grad Accumulation Steps | 1                                     |
> | Weight Decay   | 0.0                                           |
> | Label Smoothing| 0.001                                         |
> | LoRA Rank      | 256                                           |
> | LoRA Alpha     | 512                                           |
> | LoRA Dropout   | 0.05                                          |
>
> We report the results after the two modes of fine-tuning:
>
> ### COMET-22 Scores (Higher is better)
>
> |               | En-Lt (ID) | En-Lt (OOD) | En-Hi (ID) | En-Hi (OOD) | En-Ta (ID) | En-Ta (OOD) |
> |---------------|------------|-------------|------------|-------------|------------|-------------|
> | Standard Fine-tune | 0.63 | 0.70 | 0.68 | 0.58 | 0.57 | 0.61 |
> | ICA-Tune           | 0.71 | 0.75 | 0.72 | 0.63 | 0.69 | 0.76 |
>
> We observe that for both in-distribution and out-of-distribution test examples, ICA-Tune performs better than standard fine-tuning.
>
>
> (3) We repeat experiments with a reduced number of examples (14 instead of 16) in the pre-aligned sequence to ensure that the total prompt tokens match the standard version. Below, we report accuracy for the reduced pre-aligned setting. Even after equalizing the number of tokens, the pre-aligned prompting significantly outperforms standard prompting.
>
> | Model           | Standard | Pre-Aligned | Reduced Pre-Aligned |
> |----------------|----------|-------------|----------------------|
> | Llama-3.2-1B  |     27.5     |      100       |       99.38               |
> | Llama-3.2-3B  |     30.63     |      95.94       |     95                 |
> | Llama-3.2-8B  |    36.25      |     98.44        |     99.38                 |
>
> (4) For the comparison between ICA-Tune and standard fine-tuning, we use the exact same set of OOD test examples. The only difference is that, in ICA-Tune, we prepend each test example with k examples from the training set. We have mentioned this in Subsection 4.2, but we will clarify it further.

---

### Decision · Program_Chairs · 2025-05-01

**Decision:**

Accept (poster)

**Comment:**

This paper identifies a critical gap in the in-context learning capabilities of modern LLMs on seq2seq tasks and proposes a simple yet effective fine-tuning method (ICA-Tune) to address it. While initial experiments are synthetic, the authors convincingly extend their findings to real-world translation tasks and larger LLMs, addressing concerns about generalization. Reviewers appreciated the additional thorough mechanistic analysis and clarity of the contribution. Overall, this is a thoughtful and well-executed paper that makes a meaningful contribution to understanding and improving ICL for structured tasks.